# The Luteolinidin and Petunidin 3-*O*-Glucoside: A Competitive Inhibitor of Tyrosinase

**DOI:** 10.3390/molecules27175703

**Published:** 2022-09-04

**Authors:** Seo Young Yang, Jang Hoon Kim, Xiangdong Su, Jeong Ah Kim

**Affiliations:** 1Department of Pharmaceutical Engineering, Sangji University, 83 Sangjidae-gil, Wonju 26339, Korea; 2Department of Herbal Crop Research, National Institute of Horticultural and Herbal Science, RDA, Eumseong 27709, Korea; 3School of Pharmaceutical Sciences (Shenzhen), Shenzhen Campus of Sun Yat-sen University, No. 66, Gongchang Road, Guangming District, Shenzhen 518107, China; 4Vessel-Organ Interaction Research Center, VOICE (MRC), College of Pharmacy, Kyungpook National University, Daegu 41566, Korea; 5BK21 FOUR Community-Based Intelligent Novel Drug Discovery Education Unit, College of Pharmacy and Research Institute of Pharmaceutical Sciences, Kyungpook National University, Daegu 41566, Korea

**Keywords:** tyrosinase, melanin, anthocyanins, competitive inhibitor, molecular docking

## Abstract

The enzyme tyrosinase plays a key role in the early stages of melanin biosynthesis. This study evaluated the inhibitory activity of anthocyanidin (**1**) and anthocyanins (**2**–**6**) on the catalytic reaction. Of the six derivatives examined, **1**–**3** showed inhibitory activity with IC_50_ values of 3.7 ± 0.1, 10.3 ± 1.0, and 41.3 ± 3.2 μM, respectively. Based on enzyme kinetics, **1**–**3** were confirmed to be competitive inhibitors with *K*_i_ values of 2.8, 9.0, and 51.9 μM, respectively. Molecular docking analysis revealed the formation of a binary encounter complex between **1**–**3** and the tyrosinase catalytic site. Luteolinidin (**1**) and petunidin 3-*O*-glucoside (**2**) may serve as tyrosinase inhibitors to block melanin production.

## 1. Introduction

Tyrosinase (EC 1.14.18.1) is a multifunctional copper-containing enzyme that plays a role in melanin biosynthesis in mammals, plants, insects, and microorganisms [1]. It catalyzes two reactions: the hydroxylation of tyrosine to L-3,4-dihydroxyphenylalanine and the subsequent oxidation of L-3,4-dihydroxyphenylalanine to dopaquinone [2]. Melanin is produced in skin melanocytes to protect the skin from ultraviolet (UV) radiation [3]. Melanin overproduction leads to age spots, freckles, senile lentigines, solar lentigo, hyperpigmentation, and melisma [1,3], while overexposure of the skin to UV radiation causes malignant melanoma [4]. These disorders may be directly related to the tyrosinase catalytic reaction [4]. To address these problems, tyrosinase inhibitors with high activity and low toxicity have been developed from natural products [1], such as methyl hesperidin [5], broussoflavonol J [6], and dieckol [7].

Anthocyanins are polyphenols that dissolve in water [8] and are flavonoids biosynthesized from phenylalanine and malonyl-CoA along with flavanols [9]. They exist in edible plants, such as purple corn, black soybean, blueberry, and grape, and are responsible for the plants’ red, purple, and blue colors [8,9]. The carbon skeleton of anthocyanins is composed of C-6 (A-ring), C-3 (C-ring), and C-6 (B-ring) carbons [10]. Their colors depend on the amount of hydroxylation in the B-ring [10]. Their maximum absorption occurs at 450–550 nm [10]. Cyandin, delphinindin, and pelargonidin glycosides such as cyanidin 3-*O*-glucoside, delphinidin 3-*O*-glucoside, and pelargonidin 3-*O*-glucoside are the most abundant naturally occurring anthocyanins, which are widely found in *Morus nigra*, *Berberis vulgaris*, and *Ipomea batatas* [11]. Anthocyanins possess a variety of bioactivities, such as anti-inflammation [12], anti-cancer [13], and renal-protective [14] effects. Cyanidin 3-*O*-glycoside and peonidin 3-*O*-glucoside suppress matrix metalloproteinase expression in IL-1β-stimulated human articular chondrocytes [12]. Delphinidin, which is a derivative of an anthocyanin, inhibits glyoxalase 1 and is overexpressed in tumor cells [13]. Cyanidin 3-*O*-glucoside is protected against diabetic nephropathy by decreasing renal TNF-α mRNA and NF-κB mRNA levels in rats [14].

## 2. Results

### 2.1. Inhibitory Activity of Antocyanins on Tyrosinase

This study evaluated the inhibitory effects of one anthocyanidin and five anthocyanins on tyrosinase: luteolinidin (**1**), petunidin 3-*O*-glucoside (**2**), delphinidin 3-*O*-galactoside (**3**), kuromanin (cyanidin 3-*O*-glucoside) (**4**), delphinidin 3-*O*-glucoside (**5**), and callistephin (pelargonidin 3-*O*-glucoside) (**6**) (Figure 1). At 50 μM, compounds **1**–**3** had inhibitory ratios exceeding 55% of the control value (Table 1), showing dose-dependent inhibitory activity, with IC_50_ values of 3.7 ± 0.8, 10.3 ± 1.0, and 41.3 ± 3.2 μM, respectively (Figure 1 and Table 1). 

### 2.2. Enzyme Kinetics of the Compounds on Tyrosinase

To reveal the mechanism of binding between compounds **1**–**3** and the enzyme, an enzyme kinetic study was performed using various substrate concentrations. The *v*_0_ was calculated at ~10% of the substrate conversion rate by tyrosinase. The results are shown as Lineweaver–Burk plots (Figure 2A–C and Table 1). Compounds **1**–**3** all had different slopes (*K*m/*V*max) and different y-axis intercepts (*V*max), demonstrating that they bound to the enzyme reaction site competitively. Using Dixon plots, **1**–**3** were calculated to have inhibition constants (*K*_i_) of 2.8, 9.0, and 51.9 μM, respectively (Figure 2D–F and Table 1).

### 2.3. Molecular Docking of the Compounds with Tyrosinase

To determine the binding orientation of inhibitors in tyrosinase, a molecular simulation analysis was performed using AutoDock 4.2. Based on an enzyme kinetic study, a grid containing the active site with two copper ions was used to simulate the interactions of the inhibitors with the tyrosinase catalytic site. As shown in Figure 3A−C and Table 2, inhibitors **1** and **3** formed two different clusters according to the binding position. However, their Autodock scores were similar. Based on enzyme theory, competitive inhibitors bind mainly to the catalytic site. Out of 50 ranks, cluster 1 next to the catalytic site was created only by ranks 1–3 (**1**) and 1–5 (**3**) [15].

These findings suggested that cluster 2 may be the site that interacts with inhibitors **1** and **3**. As indicated in Figure 3A–C and Table 2, **1**–**3** were stably anchored in the active site of tyrosinase with respective Autodock scores of −5.36, −4.96, and −4.61 kcal/mol. Inhibitor **1** formed four hydrogen bonds with three amino acids (2.71 Å from His244, 2.65 and 2.72 Å from Glu256, and 2.53 Å from Gly281). The five hydroxyl groups of inhibitor **2** were located at distances of 2.78 Å from Asn260, 3.13 Å from Arg268, 2.94 Å from Gly281, and 2.73 Å and 3.15 Å from Ser282. Inhibitor **3** formed three hydrogen bonds: 2.83 Å from Asn260 and two 3.14 Å from Arg268.

## 3. Experimental Section

### 3.1. Chemical Reagents

Tyrosinase (T3824), kojic acid (K3125), and L-tyrosine (T3754) were purchased from Sigma–Aldrich (St. Louis, MO, USA). UV-Vis spectra were obtained using the TECAN infinite 200 PRO^®^ spectrophotometer (Zurich, Switzerland). Luteolinidin (**1**), petunidin 3-*O*-glucoside (**2**), delphinidin 3-*O*-galactoside (**3**), kuromanin (cyanidin 3-*O*-glucoside) (**4**), delphinidin 3-*O*-glucoside (**5**), and callistephin (pelargonidin 3-*O*-glucoside) (**6**) were purchased from LGC Standards (1154-78-5, Teddington Middlesex, UK) and Sigma–Aldrich (**2**, 30638; **3**, 04301; **4**, 52976; **5**, 73705; **6**, 79576; St. Louis, MO, USA).

### 3.2. Tyrosinase Assay

To assess the inhibitory effect of compounds on tyrosinase, 130 μL tyrosinase in 0.05 mM phosphate buffer (pH 6.8) was aliquoted into 96-well plates [16], and 20 μL of each compound at concentrations ranging from 0.5 to 0.0075 mM was added. Next, 50 μL 1.2 mM L-tyrosine in phosphate buffer was diluted to calculate the inhibitory activity. Finally, 50 μL 10–0.62 mM l-tyrosine in buffer was added to analyze the initial velocity (*v*_0_). Twenty minutes after starting the reaction, the amount of product was detected at 475 nm. The inhibitory activity was calculated using the following equation: Inhibitory activity rate (%) = [(ΔC − ΔS)/ΔC] × 100, where C and S are the intensity of control and inhibitor after 20 min, respectively.

### 3.3. Molecular Docking

The three-dimensional structure of the protein encoded by 2Y9X was determined from the Research Collaboratory for Structural Bioinformatics homepage. Hydrogen atoms were added to this and assigned Gasteiger charges using AutoDockTools. Ligands were built and their energy minimized by MM2 using Chem3D Pro. A grid containing the active site was established (X = 60, Y = 60, Z = 60). A DPS file was constructed to set up a Lamarckian genetic algorithm for ligand docking with the receptor (50 runs, maximum number set as long). The results were presented using Chimera (San Francisco, CA, USA) and LIGPLOT (Cambridge, UK).

### 3.4. Statistical Analysis

All inhibitory concentration data were obtained from independent experiments carried out in triplicate. Results are shown as the mean ± standard error of the mean (SEM). The results were subjected to analysis using Sigma plot 10.0 (Systat Software Inc., San Jose, CA, USA). 

## 4. Discussion

The anthocyanin-rich fractions of blueberry extract decreased the proliferation of B16-F10 melanoma murine cells by inducing apoptosis [17]. The major anthocyanins in black rice (peonidin 3-*O*-glucoside and cyanidin 3-*O*-glucoside) were confirmed to decrease MMP-2 and u-PA secretion in SCC-, Huh-7, and HeLa cells [18]. Recently, the ethyl acetate fraction from *Arctium lappa* L. leaves, which contained *trans*-5-caffeoylquic acid, rutin, kaempferol 3-*O*-rutinoside, 3,5-*di*-*O*-caffeoylquic acid, and 4,5-*di*-*O*-caffeoylqunic acid, was found to decrease tyrosinase activity and melanin levels [19]. This downregulated the expression of phosphorylated c-Jun N-terminal kinase, microphthalmia-associated transcription factor, tyrosinase-related protein 1, and tyrosinase in α-MSH-induced B16.F10 cells [19].

Luteolinidin (**1**) from *Sorghum bicolor* [20] and petunidin 3-*O*-glucoside (**2**) from *Vitis vinifera* L. [21] were the most potent inhibitors on tyrosinase, whereas delphinidin 3-*O*-galactoside [22] (**3**) showed moderate inhibitory activity. Many studies have examined the inhibitory effects of anthocyanin-rich fractions on tyrosinase [23,24], whereas we examined the tyrosinase inhibitory activity of single anthocyanins from plants. As a result, the anthocyanidin (**1**) and anthocyanins (**2**–**3**) showed the inhibitory activity within the concentration range of micromole on tyrosinase. Some flavonoids have recently been reported to interact directly with the tyrosinase catalytic site [25]. In addition, inhibitors **1**–**3,** as competitive inhibitors, were confirmed to be bound into a common active site in an in vitro assay. The molecular simulation calculated that they were anchored into the cavity hole in tyrosinase. In particular, two inhibitors, **1** and **3**, formed cluster 1 stably in the left position next to the active site. Based on enzyme kinetic theory, it can be confirmed that this cluster, formed by the two inhibitors, had a logical error. Therefore, cluster 2, formed by the two inhibitors on the catalytic site, was determined to be the predicted binding calculated by the Autodock program. According to the above results, it was confirmed that glycosides **2** and **3** form a similar Autodock score and binding poses. Inhibitors **2** and **3**, which are similar to the backbone of inhibitor **1**, were bound as different poses in the catalytic site by their sugars. 

## 5. Conclusions

Many studies have examined the tyrosinase inhibitory activity of anthocyanin extracts from plants [26]. We evaluated how these compounds inhibit the catalytic reaction. Of the one anthocyanidin and five anthocyanins evaluated, luteolinidin (**1**), petunidin 3-*O*-glucoside (**2**), and delphinidin 3-*O*-galactoside (**3**) acted as tyrosinase inhibitors at micromolar concentrations. These compounds bound competitively, with *K*_i_ values of 2.8, 9.0, and 51.9 μM, respectively. A molecular simulation of the binding of the compounds to the catalytic site of tyrosinase confirmed that inhibitors **1** and **2** interact with tyrosinase with an excellent inhibitory effect and low Autodock binding energy. Therefore, we suggest that luteolinidin (**1**) and petunidin 3-*O*-glucoside (**2**) should be the lead compounds for developing new tyrosinase inhibitors.

## Figures and Tables

**Figure 1 molecules-27-05703-f001:**
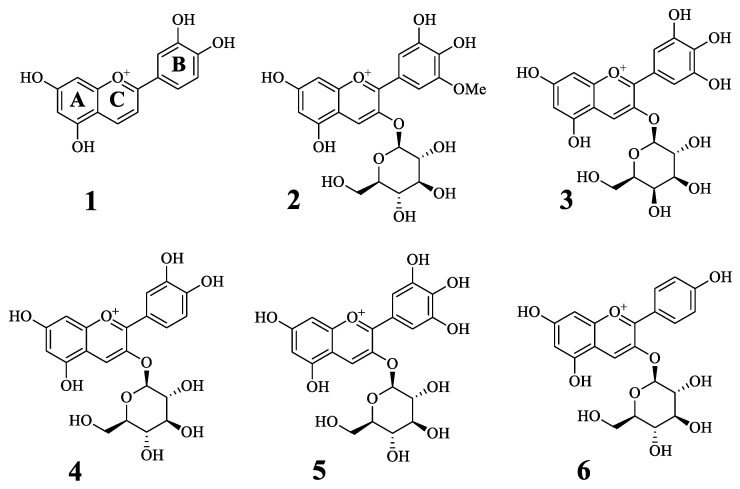
The structures of compounds **1**–**6**.

**Figure 2 molecules-27-05703-f002:**
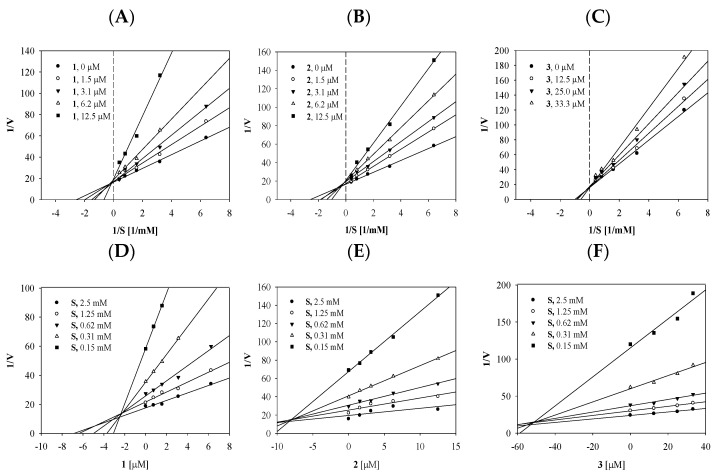
Lineweaver–Burk (**A**–**C**) and Dixon (**D**–**F**) plots of inhibitors **1**–**3**, the results were subjected to analysis using Sigma plot 10.0.

**Figure 3 molecules-27-05703-f003:**
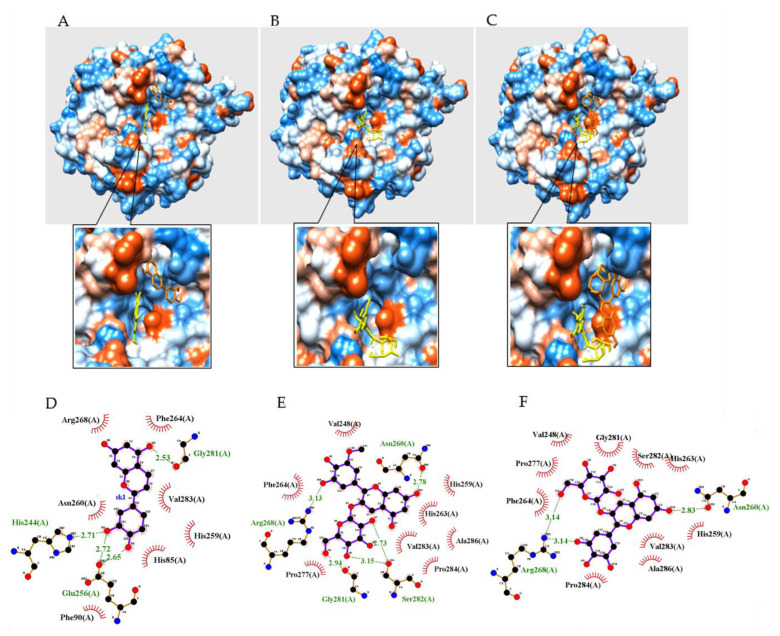
The best binding poses (**A**–**C**) of compounds **1**–**3** on tyrosinase (orange structure is best pose of cluster 1 of inhibitors **1** and **3**, and yellow structure is best pose of cluster 2 of inhibitors **1** and **3**, and cluster 1 of inhibitor **2**). The green dotted line represents hydrogen bonds’ interaction between inhibitor **1**–**3** (**D**–**F**) and enzyme, respectively.

**Table 1 molecules-27-05703-t001:** Inhibitory activity of compounds **1**–**6** on tyrosinase.

Compound ^a^	Inhibitory Ratio at 50 μM	IC_50_ (μM) ^d^	Binding Mode (*k*_i_, μM)
**1**	176.5 ± 11.6	3.7 ± 0.8	Competitive inhibitor (2.8 ± 0.5)
**2**	69.2 ± 1.0	10.3 ± 1.0	Competitive inhibitor (9.0 ± 2.1)
**3**	55.2 ± 0.9	41.3 ± 3.2	Competitive inhibitor (51.9 ± 3.8)
**4**	−1.9 ± 40.0	N.T.	N.T. ^c^
**5**	47.5 ± 3.2	N.T.	N.T. ^c^
**6**	6.5 ± 2.1	N.T.	N.T. ^c^
Kojic acid ^b^		31.4 ± 1.1	

^a^ Compounds were tested three times. ^b^ Positive control ^c^ N.T.: not test. ^d^ Mean ± SEM.

**Table 2 molecules-27-05703-t002:** Hydrogen bonds analysis of the compounds with tyrosinase.

	Autodock Energy (kcal/mol)	Hydrogen Bonds (Å)
**1**	−5.36	His244(2.71), Glu256(2.65, 2.72), Gly281(2.53)
**2**	−4.96	Asn260(2.78), Arg268(3.13), Gly281(2.94), Ser282(2.73,3.15)
**3**	−4.61	Asn260(2.83), Arg268(3.14, 3.14)

## Data Availability

Not applicable.

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
