# Peer review of "The Luteolinidin and Petunidin 3-O-Glucoside: A Competitive Inhibitor of Tyrosinase"

_molecules, 2022, doi:10.3390/molecules27175703_

Round 1

Reviewer 1 Report

The authors successfully and efficiently evaluated inhibitory activity of anthocyanins. The enzyme kinetics show luteolinidin and petunidin-3-O-glucosidase are competitive tyrosinase inhibitors and works in uM concentration level. Experimental data and methods are sufficient to conclude their argument, but data presentation can be improved. I recommend accepting this manuscript after the minor revisions as suggested below.

1.     “50 uM” at the title of second column in Table 1 is not clear. Please rewrite that as “Inhibitory ratio at 50 uM”

2.     Add statistical details in both experimental and result section.

3.     Is IC50 presented as Mean±SD or Mean±SEM. Please mention that in Table 1.

4.     Please provide error values of Ki for compound 1-3 in Table 1, last column as well as in text.

5.     Please specify the name and version of software was used for L-B plot and Dixon plot in both text and Figure legend.

6.     Figure legends needs to be more details.

7.     ‘l’- at L-tyrosine in Page 4 line 109 and 119 should be in capital.

Author Response

We deeply appreciate your help on the promotion of our revised manuscript entitled “The anthocyanins luteolinidin and petunidin-3-O-glucoside are competitive tyrosinase inhibitors”. (Manuscript ID: molecules-1877832) We have revised the manuscript and supplementary materials step by step based on the reviewers’ comments. And we changed the authorhip for our own reasons, and all the authors agreed. We would appreciate it if you could understand our situation. In addition to the professional comments, all of authors carefully read and checked both manuscript and supplementary materials and some corrections have been made and marked in yellow.

For details, please see below:

Correction highlights

The authors successfully and efficiently evaluated inhibitory activity of anthocyanins. The enzyme kinetics show luteolinidin and petunidin-3-O-glucosidase are competitive tyrosinase inhibitors and works in uM concentration level. Experimental data and methods are sufficient to conclude their argument, but data presentation can be improved. I recommend accepting this manuscript after the minor revisions as suggested below.

  1. “50 uM” at the title of second column in Table 1 is not clear. Please rewrite that as “Inhibitory ratio at 50 uM”
  • According to your kind suggestion, we revised “50 uM” to “Inhibitory ratio at 50 uM” at the title of second column in Table 1. The corrections were highlighted in the manuscript
  1. Add statistical details in both experimental and result section.
  • Thank you for this important advice, the statistical details were added in both experimental and result sections
  1. Is IC50 presented as Mean±SD or Mean±SEM. Please mention that in Table 1.
  • The IC50 presented as Mean±SEM in this manuscript. It was highlighted in Table 1
  1. Please provide error values of Ki for compound 1-3 in Table 1, last column as well as in text.
  • Accoridng to this comment, the error values of Ki for compounds 1-3 were added in the Table 1
  1. Please specify the name and version of software was used for L-B plot and Dixon plot in both text and Figure legend.
  • According to this comment, the specific name and verison of software for Linewear-Burk and Dixion plots were added in Statistical analysis section and Figure legend. Those revisions were highlighted in the manuscript
  1. Figure legends needs to be more details.
  • Thanks for your considerate suggestion. However, the details of each figures were deeply discussed in the content of discussion section
  1. ‘l’- at L-tyrosine in Page 4 line 109 and 119 should be in capital.
  • Accoridng to this comment, the ‘l’ of L-tyrosine was corrected to be in captial font. This correction were highlighted in the manuscript.

Reviewer 2 Report

The current manuscript describes the inhibitory effect of natural glycosides on tyrosinase. The work is nicely conducted and well reported. The only major concern relies on the control: why using Kojic acid and not arbutin which is more closely related to the tested molecules ? I will advise the authors to perform this experiment. The authors should also compare their molecules to others from previous studies. See for example: Eur. J. Org. Chem. 2021, 3812-3818, and many others...

Author Response

We deeply appreciate your help on the promotion of our revised manuscript entitled “The anthocyanins luteolinidin and petunidin-3-O-glucoside are competitive tyrosinase inhibitors”. (Manuscript ID: molecules-1877832) We have revised the manuscript and supplementary materials step by step based on the reviewers’ comments. And we changed the authorhip for our own reasons, and all the authors agreed. We would appreciate it if you could understand our situation. In addition to the professional comments, all of authors carefully read and checked both manuscript and supplementary materials and some corrections have been made and marked in yellow.

For details, please see below:

Correction highlights: 

The current manuscript describes the inhibitory effect of natural glycosides on tyrosinase. The work is nicely conducted and well reported. The only major concern relies on the control: why using Kojic acid and not arbutin which is more closely related to the tested molecules ? I will advise the authors to perform this experiment. The authors should also compare their molecules to others from previous studies. See for example: Eur. J. Org. Chem. 2021, 3812-3818, and many others...

  • Thank you for your kind suggestion. We evaluated the tyrosinase inhibitory activies of both kojic acid and arbutin. However, in our enzymatic exprimental system arbutin showed a very weak inhibitory activity against tyrisinase (IC50 value of arbutin: 294.2±0)

Moreover, a published paper suggested kojic acid was a suitable postitive control among other possible compounds including α-arbutin and β-arbutin due to its broad application for both vitro and vivo assays and no cytotoxity1.

[1] Wang, W.; Gao, Y.; Wang, W.; Zhang, J.; Yin, J.; Le, T.; Xue, J.; Engelhardt, U.H.; Jiang, H. Kojic Acid Showed Consistent Inhibitory Activity on Tyrosinase from Mushroom and in Cultured B16F10 Cells Compared with Arbutins. Antioxidants 202211, 502. https://doi.org/10.3390/antiox11030502

Round 2

Reviewer 2 Report

the authors have answered with satisfaction to the comments.

Author Response

We deeply appreciate your help.